# Construction of a Dengue NanoLuc Reporter Virus for In Vivo Live Imaging in Mice

**DOI:** 10.3390/v14061253

**Published:** 2022-06-09

**Authors:** Enyue Fang, Xiaohui Liu, Miao Li, Jingjing Liu, Zelun Zhang, Xinyu Liu, Xingxing Li, Wenjuan Li, Qinhua Peng, Yongxin Yu, Yuhua Li

**Affiliations:** 1Department of Arbovirus Vaccine, National Institutes for Food and Drug Control, Beijing 102629, China; dlu_fangenyue@163.com (E.F.); liux1aohui@163.com (X.L.); limiao1711@163.com (M.L.); liujingjing@nifdc.org.cn (J.L.); zelun128@hotmail.com (Z.Z.); liuxinyu@nifdc.org.cn (X.L.); lxx0320@outlook.com (X.L.); wenjuan279@outlook.com (W.L.); pengqinhua666@163.com (Q.P.); 2Wuhan Institute of Biological Products, Co., Ltd., Wuhan 430207, China

**Keywords:** flavivirus, reporter virus, in vivo imaging, neurovirulence

## Abstract

Since the first isolation in 1943, the dengue virus (DENV) has spread throughout the world, but effective antiviral drugs or vaccines are still not available. To provide a more stable reporter DENV for vaccine development and antiviral drug screening, we constructed a reporter DENV containing the NanoLuc reporter gene, which was inserted into the 5′ untranslated region and capsid junction region, enabling rapid virus rescue by in vitro ligation. In addition, we established a live imaging mouse model and found that the reporter virus maintained the neurovirulence of prototype DENV before engineering. DENV-4 exhibited dramatically increased neurovirulence following a glycosylation site-defective mutation in the envelope protein. Significant mice mortality with neurological onset symptoms was observed after intracranial infection of wild-type (WT) mice, thus providing a visualization tool for DENV virulence assessment. Using this model, DENV was detected in the intestinal tissues of WT mice after infection, suggesting that intestinal lymphoid tissues play an essential role in DENV pathogenesis.

## 1. Introduction

It has been nearly 80 years since the first dengue virus (DENV) was isolated in 1943. Four DENV serotypes are now circulating worldwide, with infection in more than 100 countries and nearly 3.6 billion people at risk of infection [1]. With the rise of reverse genetics technology, recombinant virus design has been widely used in basic research on RNA viruses and vaccine development, which has played a key role in the prevention and control of DENV and other flaviviruses [2,3]. In particular, reporter viruses are widely used as a molecular tool for antiviral drug screening [4,5], neutralizing antibody quantification [6,7,8,9,10,11], vaccine development [12], and pathogenesis research [13,14].

Originally, reporter viruses were constructed by inserting the reporter gene into a permissive site in the 3′ untranslated region (UTR) under the control of an internal ribosomal entry site [15,16,17]. Despite short-term success, these reporter viruses were prone to loss of reporter genes due to recombination, which made them unstable. Subsequently, a different and more stable method of inserting the reporter gene was developed for yellow fever virus. It is important to insert the reporter gene into the linkage region between the 5′ UTR and the capsid gene, but because of RNA regulatory elements that are continuous from the 5′ UTR to the capsid gene, a portion of the capsid gene must be duplicated upstream of the reporter gene [18]. Further improvements to achieve long-term stability (i.e., stable for ten passages in cell culture) have been developed. Besides using split reporter systems, codon optimization of the capsid gene to reduce homology and optimization of capsid duplication length can increase reporter stability [19]. Although these new approaches can improve reporter virus construction, the insertion of reporter genes into the viral genome as exogenous genes can reduce the virulence of engineered flaviviruses. Such attenuation weakens the utility of current reporter viruses when studying viral replication and pathogenesis. 

Mice are not sensitive to DENV; in the early stage, intracranial infection of wild-type (WT) mice is often used for virulence evaluation by natural survival [20,21]. The DENV Ban18 strain isolated from mosquitoes in Xishuangbanna, Yunnan, China, in 1981 has strong neurovirulence in mice. Subsequently, the Ban18HK20 strain, attenuated for mice, was obtained by successive passages on primary hamster kidney cells for 20 generations [22]. We previously found that amino acid 155 of the E protein of the Ban18 strain is a key site affecting virus virulence. Intracranial infection of BALB/c mice with a glycosylation site-defective mutation at amino acid 155 (T155I) in the envelope (E) protein of DENV-4 Ban18HK20 strain resulted in neurological morbidity and mortality in mice with significantly increased neurovirulence. However, this method does not allow the observation of the virus infection cycle in vivo, which is not conducive to the study of pathogenesis. Although a reporter DENV, based on a conventional luciferase construct, could be visualized in vivo in mice [13], a stable reporter virus still only allows the insertion of small exogenous genes. The insertion of reporter genes of proteins with large molecular weight into flaviviruses can attenuate the pathogenicity and affect the virulence mechanisms of these viruses in relevant applications [19]. In this study, we used the gene of NanoLuc luciferase, which has a low molecular weight and high luminescence intensity, to construct a reporter DENV. For the study of the pathogenesis and virulence of DENV, we also established a live imaging mouse model to provide a visualization tool for observing the dynamic process of DENV infection for a continuous period without dissecting mice.

## 2. Materials and Methods

### 2.1. Viruses, Plasmids, and Cells

The neurovirulent DENV-4 Ban18 strain was isolated in 1981 from Xishuangbanna, Yunnan, China. Following 20 consecutive passages of Ban18 in primary hamster kidney cells, the non-neurovirulent DENV-4 Ban18HK20 strain was obtained. Both strains are preserved in the Division of Arboviral Vaccines, National Institutes for Food and Drug Control (NIFDC), China. Infectious clone plasmids pSPTM-DENV(WT) and pSPTM-DENV(T155I) against the DENV-4 Ban18HK20 strain and its E protein amino acid mutant strain (with mutation at position 155), as well as their rescued viruses, are preserved in the Division of Arboviral Vaccines, NIFDC. Vero cells were derived from ATCC and cultured in DMEM supplemented with 10% fetal bovine serum. Four-week-old BALB/c mice without specific pathogens were supplied by the Center of Animal Breeding, NIFDC.

### 2.2. Molecular Cloning Strategy

The NanoLuc reporter gene was derived from the commercial plasmid pNL1.1[NLuc] Vector (Cat. N1001; Promega, Madison, WI, USA). To make NanoLuc more stable when inserted into the viral genome as an exogenous gene, it was inserted following the coding sequence of the first 38 amino acids from the 5′ end of the DENV capsid gene, connected by a self-cleaving peptide coding sequence of Thosea asigna virus 2A (T2A). In addition, the coding sequence of the first 38 amino acids of the codon-optimized capsid gene (C38) was inserted following the T2A sequence to prevent the loss of the NanoLuc reporter gene by homologous recombination of the two C38 sequences.

Because of the presence of the DENV gene, which is toxic to bacteria, in the target sequence, the plasmid of the recombinant reporter virus cannot be stably replicated in *E. coli*. Therefore, the full length of the target sequence was divided into two sequences (A and B) to construct subcloned plasmids, and then the virus was rescued by enzymatic digestion and in vitro ligation of the two linear fragments (Figure 1B).

Briefly, first, the NanoLuc+T2A+C38_codon scrambling_ gene (690 bp) was synthesized. The subcloned plasmid pGS1-NLuc containing the genes of pSPTM-DENV(WT)-AscI-terminal sequence (1–241 bp), NanoLuc+T2A+C38_codon scrambling_ sequence (690 bp), and pSPTM-DENV(WT)-Bsu36I-terminal sequence (242–657 bp) was constructed using the In-Fusion Cloning Kit (Cat. 639650; TaKaRa Bio Inc., Kusatsu, Shiga, Japan). The plasmid was digested with AscI and Bsu36I, followed by treatment with calf intestinal alkaline phosphatase (CIAP), and the target sequence A (1336 bp) was obtained by electrophoresis and gel recovery. Subsequently, the pSPTM-DENV(WT) infectious clone plasmid was digested with XhoI, the sticky ends were digested with mung bean nuclease, and the linearized DNA was purified after treatment with CIAP. The recovered DNA was then digested with Bsu36I, and the target sequence B (10,043 bp) was obtained by electrophoresis, followed by gel cutting and recovery. Finally, sequences A and B were ligated at a molar concentration ratio of 3:1 in the presence of T4 ligase to obtain a full-length cDNA sequence. After electrophoresis, the ligated products were excised from the gel and used as transcriptional templates for subsequent experiments.

### 2.3. In Vitro Transcription and Virus Rescue

Full-length linear DNA obtained by in vitro ligation was used as a template for in vitro transcription using SP6 Transcription Reagent (Cat. P1280; Promega) and m7G Cap Analog (Cat. P1712; Promega). RNA was purified using the RNeasy MinElute Cleanup Kit (Cat. 74204; Qiagen, Valencia, CA, USA), and the recovered RNA was electrotransfected into Vero cells by the Gene Pulser Xcell electroporation system (Bio-Rad, Hercules, CA, USA) using the following settings: voltage: 220 V, capacitor: 300 μF, cuvette gap: 0.4 cm, resistor: none. Obvious cytopathic effects were observed after 5–7 days of cell culture, and the cell supernatant was collected as the rescued virus, centrifuged, and aliquoted for freezing at −80 °C.

### 2.4. Virus Titer Assay

Vero cells were grown in a six-well plate with 1 × 10^6^ cells/well. When the confluence reached 80–90%, the supernatant was discarded, and the virus at 10-fold serial dilution (10^−1^–10^−6^) was added to the six-well plate and incubated at 37 °C for 1 h. Excess virus was discarded, methylcellulose was added, and incubation was continued for 7 days. Plaques were counted after crystal violet staining. One plaque was considered one plaque forming unit (PFU), and the virus titer was defined as lg (PFU/mL).

### 2.5. NLuc Expression in Vero Cells

Vero cells were infected with DENV(WT)-NLuc and DENV(T155I)-NLuc at multiplicity of infection (MOI) of 0.1, and a blank control was established with cells not infected with the virus. Intracellular fluorescence expression was detected at 5 DAI when significant cytopathic effects could be observed. The substrate was diluted 1:50 with phosphate-buffered saline (PBS) in Nano-Glo^®^ Luciferase Assay (Cat. N1110; Promega), mixed with cells in equal volume, and placed in the GloMax 96 microplate luminometer (Promega) for detection.

### 2.6. RT-PCR and Viral Genome Sequencing

Viral RNA was extracted and reverse transcribed into cDNA. Primers were designed for the 5′ and 3′ ends of the NLuc gene for RT-PCR, and the amplified products were electrophoresed to observe whether the band sizes were correct. Primers were also designed for the complete reporter DENV gene sequence. The whole length of the viral genome was amplified by segmentation, and the amplification products were sequenced.

### 2.7. Virus Plaque Purification Assay

The virus to be purified was inoculated into Vero cells after 10-fold serial dilution (10^−1^–10^−6^) and adsorbed at 37 °C for 1 h. The virus in the plate was discarded, and the first layer of agar overlay was added and incubated at 37 °C for 5 days. Subsequently, a second layer of agar overlay containing neutral red staining solution was added, and incubation was continued for 24 h. The monoclonal virus strain was selected under light and cultured in a six-well plate containing Vero cells.

### 2.8. In Vivo Imaging and Dynamics in Mice

Neurovirulence and neuroinvasiveness of the recombinant reporter virus were detected in 4-week-old (13–15 g) BALB/c mice. The neurovirulence group was injected intracerebrally with 0.03 mL of virus (4.4 lg PFU). The neuroinvasiveness group was injected subcutaneously with 0.1 mL of virus (5.0 lg PFU) and intracerebrally (right side) with PBS. Prototype DENV without the reporter gene was used as a control. Fluorescence was detected in the same mice from each group at 0 (12 h), 3, 5, 7, and 9 DAI. The heart, liver, spleen, lung, kidney, brain, and intestinal tissues of mice were dissected at 7 DAI, and the fluorescence intensity was measured. For in vivo imaging of mice, the Nano-Glo substrate was diluted 1:40 with PBS and injected into the orbit. After 10 min of administration, mice or tissue organs were placed in the IVIS^®^ Spectrum in vivo imaging system (Perkin-Elmer, Waltham, MA, USA) to detect luminescence intensity. In addition, five mice in each group were infected in the same manner, and their body weight changes and survival were monitored daily.

### 2.9. Statistical Analysis

Statistical analysis was performed using GraphPad Prism 9 (GraphPad Software Inc., San Diego, CA, USA). One-way analysis of variance (ANOVA) was used to determine statistical differences between groups in luciferase activity of Vero cells after reporter DENV infection, and Tukey’s method was used for multiple comparisons. Statistical significance of the fluorescence signals in mice infected with the reporter virus was determined using two-way ANOVA. *p* < 0.05 was considered significant for differences between sample groups.

## 3. Results

### 3.1. Construction and Biological Properties of Reporter DENV

To construct a more stable reporter DENV, the NanoLuc reporter gene was engineered at the capsid gene region, and the capsid gene sequence was duplicated to flank the reporter gene (Figure 1A). Such duplication of the coding sequence of the 38 amino acids in the capsid gene (C38) is essential for viral replication, and the beginning of the capsid gene was codon scrambled to minimize homology with the duplicated portion [23]. The reporter gene was linked to the capsid gene by the T2A self-cleaving peptide sequence, and a glycine-serine-glycine (GSG) linker was added to the N-terminus of the 2A peptide sequence to improve the efficiency of the 2A peptide-induced cleavage [24]. Because DENV genes are toxic to bacteria, rapidly constructing a stable full-length cDNA infectious clone is difficult. Therefore, a full-length cDNA template for transcription was constructed by ligation of two linear fragments in vitro, and the recombinant virus was rescued by electrotransfection. Thus, two types of recombinant reporter DENVs were constructed: (i) DENV(WT)-NLuc, based on the WT DENV-4 Ban18HK20 strain; and (ii) DENV(T155I)-NLuc, based on the mutant DENV-4 Ban18HK20 strain with the glycosylation site-defective mutation (Figure 1B).

The full-length cDNA sequence of the reporter virus was obtained in vitro by ligation with T4 DNA ligase. The electrophoresis results of the purified ligated product are shown in Figure 2A, with molecular weight above 10 kb and correct band size, which was used as a template for in vitro transcription in subsequent virus rescue. The two strains of reporter DENV were titered at 6.23 lg PFU/mL for DENV(WT)-NLuc and 6.30 lg PFU/mL for DENV(T155I)-NLuc, and both showed similar plaque size (Figure 2B). Significantly higher fluorescence activity was detected after infection of Vero cells by both reporter viruses compared with the blank control (Figure 2C).

### 3.2. Plaque Purification and Sequencing of Reporter DENV

Using prototype DENV without the NanoLuc reporter gene as a control, RNA from the reporter and control viruses was extracted, and primers were designed upstream and downstream of the reporter gene region for reverse transcription-polymerase chain reaction (RT-PCR) amplification. The amplified bands of the two recombinant reporter DENVs were of the correct size but abnormal (Figure 3A), suggesting that the recombinant viruses obtained using this method were impure and required to be purified by plaque selection. Therefore, five purified monoclonal strains of each of the two reporter viruses were isolated and cultured. RT-PCR results showed that DENV(WT)-NLuc clones 2–5 and DENV(T155I)-NLuc clones 1–5 were amplified with bands of the correct size (Figure 3B). Viral genome sequencing results showed that the purified monoclonal strains 1, 2, 3, and 4 of DENV(WT)-NLuc contained nucleotide mutations (Table 1), whereas the other purified strains contained non-nucleotide mutations and could be used for subsequent study.

### 3.3. Characterization of In Vivo Propagation of Reporter DENV

DENV(WT)-NLuc and DENV(T155I)-NLuc intracerebrally injected mice were imaged in vivo on different days after infection (DAI). DENV(WT)-NLuc fluorescence was undetectable at 7 DAI, and slight fluorescence was detected in the brain at 9 DAI, whereas DENV(T155I)-NLuc was detected in the brain at 3 DAI, and the fluorescence intensity in the brain continued to increase (Figure 4). In addition, fluorescence of DENV(T155I)-NLuc was detected in the mouse peritoneal cavity at 7 DAI (Figure 4), and dissection of organs for fluorescence imaging showed that fluorescence was detectable in mouse intestinal tissues (Figure 5). Subsequently, the reporter virus in mouse intestinal tissues was transient in repeated trials.

### 3.4. Similar Neurovirulence in Reporter and Prototype DENVs

Generally, the insertion of exogenous genes into the viral genome can reduce virus virulence. To determine whether the virulence of the constructed reporter DENV is altered in mice, we injected mice with the recombinant reporter virus and prototype virus (without the reporter gene) by two routes of infection—intracerebral and subcutaneous, followed by intracerebral blank injection. The results of the intracerebral infection group showed that DENV(T155I)-NLuc had reduced neurovirulence, and the mortality rate of mice was reduced from 100% to 80% compared with DENV(T155I) that was not inserted with the NanoLuc reporter gene, but both groups had a significant reduction in body weight (Figure 6 and Figure 7). The results of the subcutaneous infection group showed that neither the reporter virus nor the prototype virus was neuroinvasive in mice, and no fluorescence was detected. The recombinant reporter virus had properties similar to the prototype virus, and the insertion of the reporter gene could retain the high DENV neurovirulence and the same symptoms of mice pathogenesis as the prototype virus, indicating that this model could be used for studies related to DENV virulence mechanism and antiviral drug screening.

## 4. Discussion

Ever since the first reporter flavivirus was reported in 2003 [15], many types of reporter viruses have been widely used in high-throughput antiviral drug screening [4], host–virus pathogenesis studies [25], and serological diagnosis [9]. Despite their several advantages, reporter flaviviruses still face the challenges of genetic instability and susceptibility to recombination-mediated loss of reporter genes during virus passaging [16].

To analyze the pathogenic mechanism of highly neurovirulent mutant strains of DENV and the infection of attenuated strains in mice, we constructed reporter viruses to perform live imaging in mice to observe the tropism and dynamic distribution of DENV infection. However, because the conventional fluorescent reporter gene is large, its insertion into the viral genome adversely affects the genetic stability and virulence of the virus, making its application difficult in animal model studies of virulence. Therefore, NanoLuc, a luciferase recently developed by Promega, was used in this study. With a low molecular weight (19.1 kDa, 171 amino acids), NanoLuc can be used for enhanced viral delivery and protein fusion, is easily secreted by cells, and is widely used in reporter flavivirus construction [7,23,26,27,28,29]. Moreover, the NanoLuc system displayed a specific activity over 150-fold higher than both North American Firefly (FLuc) and Renilla (RLuc) luciferases, providing better performance in cells that are difficult to transfect [30,31]. Although NanoLuc luciferase has a low molecular weight, it may still encounter problems with replication stability and expression or secretion as a heterologous protein when its gene is inserted directly into the DENV genome. To avoid these problems, the reporter gene was inserted between the 5′ UTR and the capsid gene. The initial amino acid-coding sequence of the capsid gene (C38) was added to the 5′ end of the reporter gene, and the T2A peptide self-cleavage sequence was added to the 3′ end of the reporter gene to enable expression and secretion of the reporter gene. Baker et al. [7,23] used a similar strategy to construct various reporter flaviviruses.

Currently, the four main 2A peptides commonly used for genetic engineering are T2A (Thosea asigna virus 2A), P2A (porcine teschovirus-1 2A), E2A (equine rhinitis A virus), and F2A (foot-and-mouth disease virus) [32,33]. Among these, T2A, P2A, and E2A can achieve almost complete self-cleaving efficiency without detection of polyprotein products, while F2A can achieve a self-cleaving efficiency of more than 90% [24,34,35,36]. In this study, the T2A peptide self-cleavage sequence was used, and the GSG sequence was added to the N-terminus of the 2A peptide sequence to improve the 2A peptide-induced cleavage efficiency [24]. Finally, to prevent homologous recombination between the C38 sequence added to the 5′ end of the reporter gene and the capsid gene sequence following T2A, the coding sequence of the first 38 amino acids of the capsid gene was codon optimized, thereby disrupting its nucleotide sequence to avoid homologous recombination and improving the stability of the recombinant reporter virus.

Based on this construction strategy, we initially attempted to construct an infectious clone plasmid by inserting the NanoLuc reporter gene directly into the DENV genome. However, because of the presence of toxicity genes in the flavivirus genome, it is difficult to directly construct infectious clone plasmids containing full-length viral sequences for stable replication in *Escherichia coli* [37,38]. Stable infectious clone plasmids with full-length viral sequences can be obtained by screening different cloning vectors and competent cells, and optimizing culture temperature, incubation time, antibiotic concentration, and shaking speed [39,40,41]. However, because this approach is time-consuming and laborious, the full length of the viral genome can be divided into multiple sequences for the subclonal construction of flavivirus infectious clone plasmids that are difficult to construct. The enzymatic cleavage sites (e.g., BsmBI and BglI) with opposite orientations (preventing self-ligation) were added to the beginning and end of the target gene of each subclone, followed by in vitro ligation of the linear fragments from multiple subcloned plasmids into the full length of viral cDNA using DNA ligase, which serves as a template for in vitro transcription in virus rescue. Deng et al. [42] rescued Zika virus by such in vitro ligation. Messer et al. [43] also successfully constructed DENV-3 and its chimeric virus using this method. Xie et al. [44,45] also used this method to rapidly construct recombinant severe acute respiratory syndrome coronavirus 2 (SARS-CoV-2) and its reporter virus, which has a larger genome than flavivirus. In this study, we also used the in vitro ligation method for virus rescue and successfully obtained reporter DENV. However, the virus rescued by in vitro ligation is usually of poor purity, and plaque purification is required for screening to obtain a correctly sequenced and genetically stable virus strain.

In this study, two reporter DENVs were constructed: (i) DENV(WT)-NLuc, a reporter virus based on the WT DENV-4 Ban18HK20 strain, which was not neurovirulent in mice; and (ii) DENV(T155I)-NLuc, a reporter virus based on the E protein N-glycosylation site-defective mutation in the mutant strain (T155I) of DENV-4 Ban18HK20, which was highly neurovirulent in mice. We previously found that the virulence of the point mutant virus DENV (T155I) was significantly enhanced after mutating the glycosylation site at amino acid 155 of the E protein of the non-neurovirulent DENV-4 Ban18HK20 strain. The same virulence locus was found in the DENV-4 H241 international standard strain by Kawano et al. [46]. Both virulent and attenuated strains containing the reporter gene were used in the subsequent in vivo live imaging study in mice. The luciferase signal was detected in the brain of DENV(T155I)-NLuc-treated mice at 3 DAI, and the expression increased day by day. The mice showed significant neurological symptoms at 7 DAI, and luciferase signals that were observed in the peritoneal cavity of mice were also detected in the intestinal tissues after dissection. Schoggins et al. [13] also detected a strong luciferase signal in gut-associated lymphoid tissue after infection of interferon receptor-deficient mice using reporter DENV. Zellweger et al. [47] detected viral RNA in the small intestine of AG129 mice after DENV-2 infection. They performed antibody-dependent enhancement experiments in which a high viral load was detected in mice in the presence of subneutralizing levels of DENV-specific antibodies. Moreover, significant gastrointestinal bleeding was observed in mice 84 h after infection, which is an important clinical hallmark of dengue hemorrhagic fever and dengue shock syndrome in humans [48]. In this study, DENV could be detected in the intestinal tissues of WT mice after infection. However, no luciferase signal was observed in the mouse intestinal tissues at 9 DAI, and subsequent retests showed that the luciferase signal in the mouse intestinal tissues was transient. Thus, intestinal lymphoid tissues may play an essential role in DENV pathogenesis. In addition, a slight luciferase signal was observed in the brain of DENV(WT)-NLuc-treated mice with no signs of morbidity at 9 DAI. Thus, the recombinant reporter virus had similar properties and virulence to prototype DENV without the reporter gene, indicating that the live imaging mouse model can be used for studies related to the virulence mechanism and antiviral drug screening of DENV.

## 5. Conclusions

NanoLuc luciferase has the twin advantages of high luminescence intensity and low molecular weight. In this study, the NanoLuc reporter gene was successfully inserted into the DENV genome to construct reporter DENVs DENV(WT)-NLuc and DENV(T155I)-NLuc based on the attenuated and virulent strains, respectively. Moreover, an in vivo imaging mouse model of intracerebral infection, which maintained DENV neurovirulence in mice, was established, thereby providing a visualization tool for the pathogenesis and antiviral drug screening of DENV. Using this model, DENV could be detected in the intestinal tissues of WT mice after infection, indicating that intestinal lymphoid tissues may play an important role in DENV pathogenesis.

## Figures and Tables

**Figure 1 viruses-14-01253-f001:**
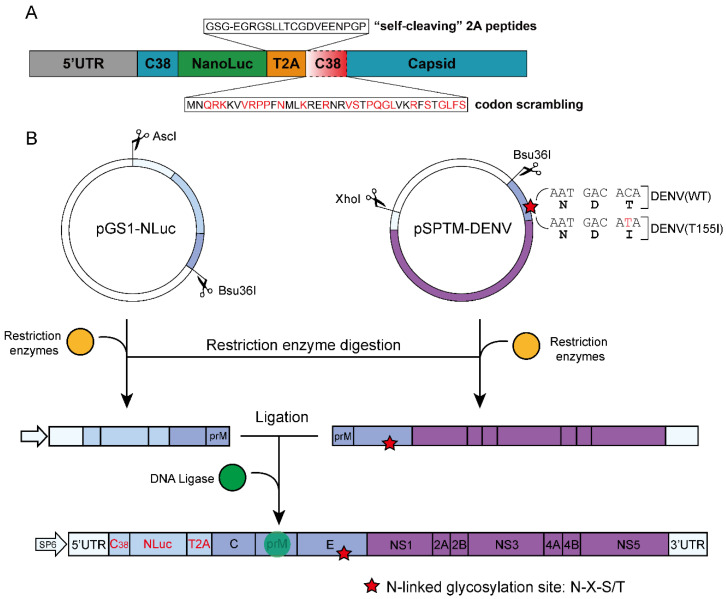
Construction of reporter DENV. (**A**) Construction design of the NanoLuc reporter gene. NanoLuc was linked to the capsid gene by the T2A sequence and was flanked by duplicate sequences of the capsid gene. (**B**) In vitro ligation of full-length cDNA of reporter DENV. Two reporter viruses were constructed, one based on WT DENV, and the other based on a glycosylation site-defective mutation in the E protein.

**Figure 2 viruses-14-01253-f002:**
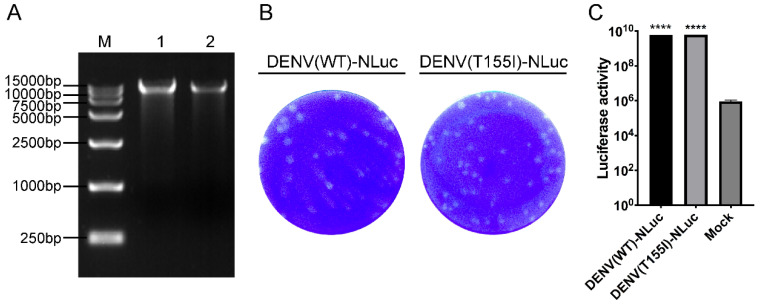
Construction and biological properties of DENV(WT)-NLuc and DENV(T155I)-NLuc. (**A**) DENV-NLuc full-length linear sequence ligated in vitro with T4 DNA ligase. M: DL15000 marker; 1: DENV(WT)-NLuc linkage products; 2: DENV(T155I)-NLuc linkage products. (**B**) Plaque morphology of DENV with the NanoLuc reporter gene. The dilution factor was 10^−4^-fold. (**C**) Luciferase activity of Vero cells after infection of reporter DENV. ****, *p* ≤ 0.0001 compared with Vero cells in the mock group.

**Figure 3 viruses-14-01253-f003:**
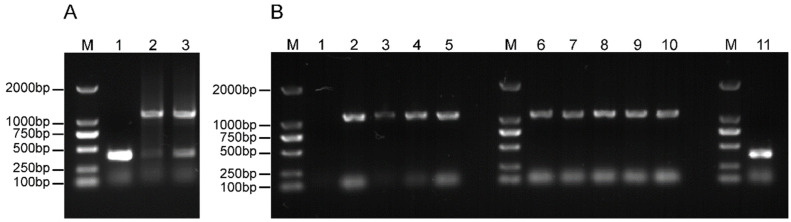
RT-PCR electrophoresis of reporter DENV. (**A**) RT-PCR of impure reporter DENV. Primers were designed to start at the 5′ UTR and end at the capsid gene, and the amplified fragment size was 1094 bp. The inserted NLuc fragment size was 690 bp. M: DL2000 marker; 1: DENV(WT); 2: DENV(WT)-NLuc; 3: DENV(T155I)-NLuc. (**B**) RT-PCR of plaque-purified reporter DENV. The amplified fragment of the virus with the NanoLuc reporter gene was 1094 bp, and the amplified fragment of the virus without the NanoLuc reporter gene was 690 bp. M: DL2000 marker; 1: DENV(WT)-NLuc-puri1; 2: DENV(WT)-NLuc-puri2; 3: DENV(WT)-NLuc-puri3; 4: DENV(WT)-NLuc-puri4; 5: DENV(WT)-NLuc-puri5; 6: DENV(T155I)-NLuc-puri1; 7: DENV(T155I)-NLuc-puri2; 8: DENV(T155I)-NLuc-puri3; 9: DENV(T155I)-NLuc-puri4; 10: DENV(T155I)-NLuc-puri5; 11: DENV(WT).

**Figure 4 viruses-14-01253-f004:**
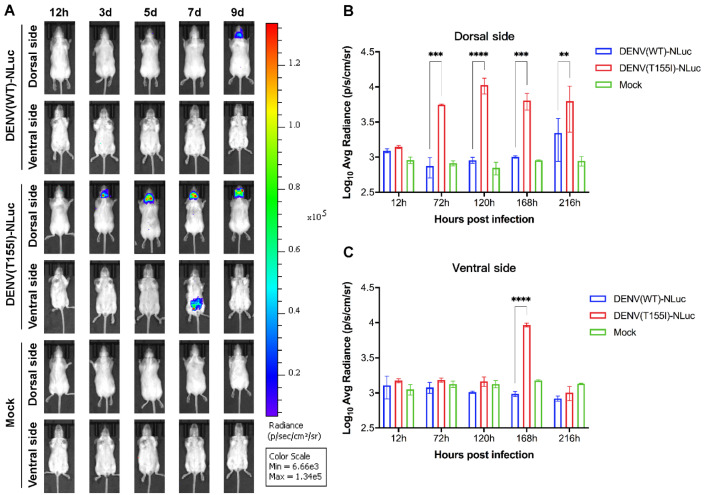
In vivo imaging of reporter DENV intracerebral infection of 4-week-old BALB/c mice. (**A**) BALB/c mice (n = 3) were injected intracerebrally with 4.4 lg PFU of the reporter virus, and fluorescence signals of the dorsal and ventral sides of mice at different time points. A representative animal from each group is shown. (**B**) Average luminescence intensity of the dorsal side of mice at different time points. (**C**) Average luminescence intensity of the ventral side of mice at different time points. Data are derived from two independent experiments. Statistical significance was determined using two-way ANOVA. **, ***, and **** indicate *p* ≤ 0.01, 0.001, and 0.0001, respectively, compared with the fluorescence signals in the DENV(WT)-NLuc group.

**Figure 5 viruses-14-01253-f005:**
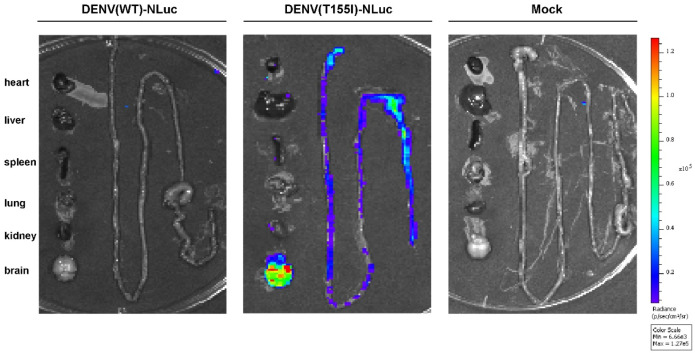
Fluorescence of tissues from major organs of mice on day 7 after intracerebral infection. BALB/c mice (n = 3) were injected intracerebrally with a dose of 4.4 lg PFU of virus, and heart, liver, spleen, lung, kidney, brain, and intestinal tissues were dissected for fluorescence detection at 7 DAI. Independent experiments were performed twice. Representative animal tissues from organs of each group are shown.

**Figure 6 viruses-14-01253-f006:**
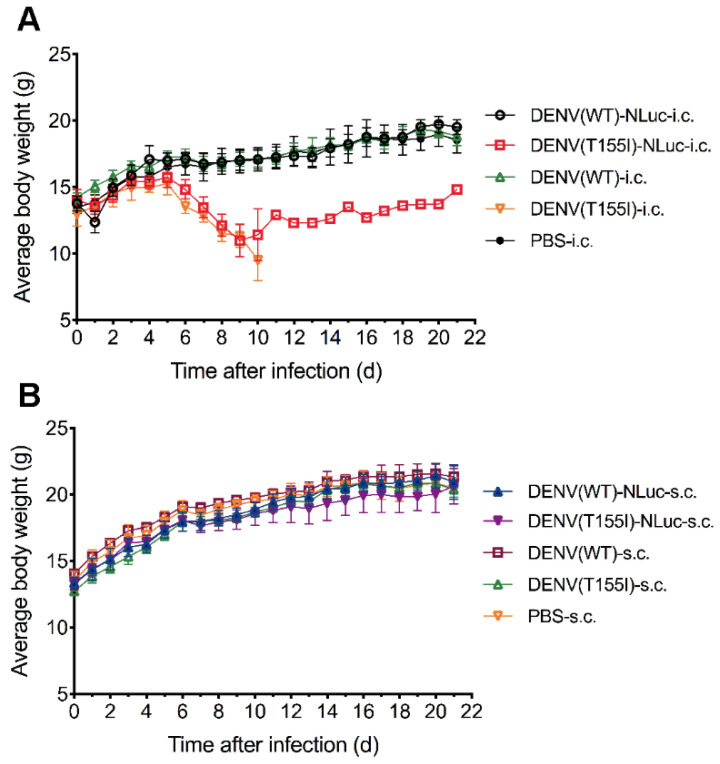
Body weight monitoring of 4-week-old BALB/c mice injected with prototype and reporter DENVs. (**A**) BALB/c mice (n = 5) were injected intracerebrally with 0.03 mL of virus (4.4 lg PFU). (**B**) BALB/c mice (n = 5) were injected subcutaneously with 0.1 mL of virus (5.0 lg PFU) and intracerebrally (right side) with PBS. Body weight of mice was monitored daily.

**Figure 7 viruses-14-01253-f007:**
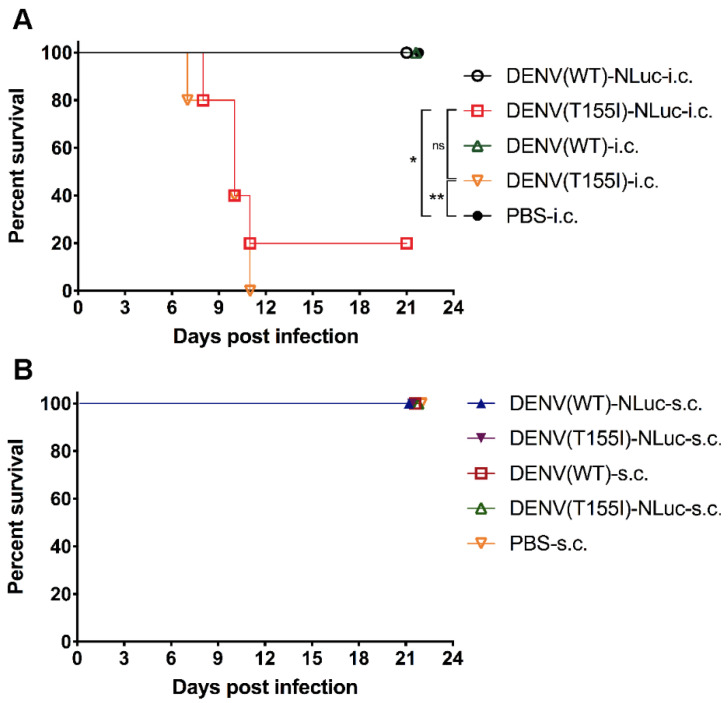
Survival curve of 4-week-old BALB/c mice injected with prototype and reporter DENVs. (**A**) BALB/c mice (n = 5) were injected intracerebrally with 0.03 mL of virus (4.4 lg PFU). (**B**) BALB/c mice (n = 5) were injected subcutaneously with 0.1 mL of virus (5.0 lg PFU) and intracerebrally (right side) with PBS. Animals were observed for 3 weeks after challenge and were euthanized when humane endpoints were reached. Log-rank (Mantel–Cox) survival analysis test was performed to determine statistical significance. *, *p* ≤ 0.05; **, *p* ≤ 0.01; ns, no statistical significance.

**Table 1 viruses-14-01253-t001:** Sequencing of plaque-purified strains of DENV with the NanoLuc reporter gene.

	Mutation	Virus Nucleotide Changes (Amino Acid Changes)											
Sample		NLuc-82	NLuc-118	T2A-14	C-1	C-15	E-224	E-241	NS1-213	NS3-14	NS3-230	NS4B-5	NS5-689
DENV(WT)-NLuc		GTG(V)	GGC(G)	GAA(E)	ATG(M)	CTG(L)	GCA(A)	AAG(K)	GAG(E)	CAG(Q)	GAG(E)	CTG(L)	AGG(R)
DENV(WT)-NLuc-puri1		-	aGC(S)	-	-	-	GCt(A)	-	-	-	-	-	-
DENV(WT)-NLuc-puri2		cTG(L)	-	-	-	CTa(L)	-	AAa(K)	-	-	GAa(E)	CTa(L)	AGa(R)
DENV(WT)-NLuc-puri3		-	-	aAA(K)	-	-	-	-	GAa(E)	CAa(Q)	-	-	-
DENV(WT)-NLuc-puri4		-	-	-	ATa(I)	-	-	-	-	-	-	-	-
DENV(WT)-NLuc-puri5		-	-	-	-	-	-	-	-	-	-	-	-
DENV(T155I)-NLuc-puri1		-	-	-	-	-	-	-	-	-	-	-	-
DENV(T155I)-NLuc-puri2		-	-	-	-	-	-	-	-	-	-	-	-
DENV(T155I)-NLuc-puri3		-	-	-	-	-	-	-	-	-	-	-	-
DENV(T155I)-NLuc-puri4		-	-	-	-	-	-	-	-	-	-	-	-
DENV(T155I)-NLuc-puri5		-	-	-	-	-	-	-	-	-	-	-	-

## Data Availability

All data are available from the corresponding authors upon request.

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
