# Peer review of "Construction of a Dengue NanoLuc Reporter Virus for In Vivo Live Imaging in Mice"

_viruses, 2022, doi:10.3390/v14061253_

Round 1

Reviewer 1 Report

In the attached manuscript, the authors describe development of a novel NanoLuc reporter dengue virus used for in vivo murine models. The authors review the development of a DENV4 clone with a NanoLuc reporter inserted prior to the capsid gene with a redundant upstream capsid sequence to maintain gene expression elements in the capsid that are required for 5’UTR function. The authors plaque isolate the reporter viruses, sequence, and characterize infection in a murine model of dengue virus infection. The work is important as it provides a novel approach to study in vivo pathogenesis of dengue virus using a reporter virus. The cloning approaches and characterization of the selected reporter virus are well explained and logically developed in the manuscript. There are some weaknesses which include a lack of specific information on experimental replicates, no clear statistical analysis in the figures that evaluate differences in pathogenesis, and lack of information on viral growth in culture and reversions or mutations that develop following multiple rounds of replication.

Major points:
Figure 4 – the results showing signal with the T155I reporter virus are important. This figure needs additional information including signal intensity values, number of replicate animals and experiments, and differences in intensity between the T155I reporter and WT reporter virus.

Figure 5- Similar to above, the finding of signal in the intestine is important. However, this image needs more information on signal intensity comparisons between groups, number of replicates, and analysis of differences in signal.

Figure 6 and 7 – these figures are busy and a bit difficult to follow. There seems to be attenuation based on body weight and survival when the NanoLuc is added in the clone, especially for the T155I virus. Consider breaking into two graphs that compare NanoLuc vs WT vs PBS for the WT virus and the T155I virus. Similar to above, need more information on replicates and statistical analysis to understand actual differences in attenuation.

Reviewer 2 Report

The authors construct reporter DENVs DENV(WT)-NLuc and DENV(T155I)- NLuc based on the attenuated and virulent strains, respectively. And they study in vivo imaging on mouse model by intracerebral infection.

1. Table 1,  the sequencing of Full-length genome of the two reporter virus is needed.

2. It is necessary to compare and study the in vivo imaging effects of foot pad and intraperitoneal injection.

3. The research background of T155I mutation of dengue virus  is unclear.Please introduce it in the introduction and discussion.

Round 2

Reviewer 1 Report

The authors have responded to all of the reviewer's comments. No additional comments on the revised manuscript.

Reviewer 2 Report

We have no other comments.